# Characterization of Coffee Silverskin from Different Origins to Evaluate Its Potential as an Ingredient in Novel Food Products

**DOI:** 10.3390/foods15010097

**Published:** 2025-12-29

**Authors:** Laura Candela-Salvador, Raquel Lucas-González, José A. Pérez-Álvarez, Juana Fernández-López, Manuel Viuda-Martos

**Affiliations:** Research Group in Innovations in Food Products (IPOA), Institute for Agri-Food and Agri-Environmental Research and Innovation, Miguel Hernández University (CIAGRO-UMH), Ctra. Beniel Km 3.2, 03312 Orihuela, Alicante, Spain; laura.candela03@goumh.umh.es (L.C.-S.); raquel.lucasg@umh.es (R.L.-G.); ja.perez@umh.es (J.A.P.-Á.); mviuda@umh.es (M.V.-M.)

**Keywords:** silverskin, dietary fiber, co-products, bioactive compounds, caffeine, polyphenols

## Abstract

Coffee silverskin is generated in large quantities as a co-product during the roasting process of coffee beans. This co-product is rich in bioactive compounds that offer potential health benefits, justifying its consideration as a functional ingredient in food. In this study, silverskin from the species *Coffea arabica* and *Coffea canephora* from six different countries was characterized to highlight its potential and applicability as a safe ingredient in new food formulations. The results revealed a dietary fiber content ranging from 71.81 to 76.86 g/100 g, with a high portion of insoluble fiber ranging from 54.02 to 60.58 g/100 g. The mineral content showed that, in all samples, potassium and calcium were the main elements with values ranging from 6.66 to 17.57 mg/g and from 9.25 to 16.44 mg/g, respectively. The caffeine content was quantified with levels ranging from 0.81 to 7.32 mg/g. In addition, high levels of phenolic compounds were identified in free and bound forms, with 5-caffeoylquinic, 3-caffeoylquinic, 4,5-dicaffeoylquinic, and ferulic acids being the main components in both fractions. All samples analyzed showed a good antioxidant capacity in the four different methods used, with values ranging from 8.12 to 10.85 mg Trolox Equivalents (mgTE/g) in the DPPH assay; from 9.69 to 19.68 mgTE/g in the FRAP assay; from 5.96 to 11.05 mgTE/g in the FRAP assay; and from 0.21 to 1.11 and 4.69 mg EDTA/g sample in the FIC assay. In conclusion, coffee silverskin has the potential to play a beneficial role as an ingredient in new food formulations, thus contributing to the development of a circular economy in the food industry.

## 1. Introduction

Coffee plants are perennial shrubs belonging to the Rubiaceae family. Although more than 120 wild coffee species have been identified, only two are commercially cultivated for global trade and consumption: *Coffea arabica* (*Arabica*) and *Coffea canephora* (*Robusta*). *C. arabica* is the most widely cultivated, representing around 55–60% of the world’s coffee supply [1]. The coffee fruits, named cherries, which are approximately 10 mm in size and oval-shaped, contain green coffee beans. These beans are enveloped by several layers: a thin skin referred to as the coffee silverskin, an endocarp known as the parchment, a peptic adhesive layer, the pulp, and the outer skin known as the epicarp [2].

Coffee is one of the most consumed beverages worldwide, mainly due to its stimulating properties [3]. It is worth noting that several studies have demonstrated that coffee has health benefits due to its high levels of antioxidants, particularly phenolic compounds, which are associated with a reduced risk of developing several chronic diseases like cancer and cardiovascular disease [4]. In addition, coffee consumption has been linked to improvements in gut microbiota and a decreased risk of diabetes, obesity, and cardiovascular mortality [5].

According to the International Coffee Organization (ICO), the estimated total world production of coffee in the year 2021–2022 was 10 million tons, while consumption of the beverage increased by 3.3%, showing steady growth over the last few years [6]. The highest production, in 2024–2025, was located in Brazil, followed by Vietnam, Colombia, and Indonesia [7]. Coffee is prepared for consumption by roasting the coffee beans at high temperatures before grinding them, to obtain a final product with a characteristic aroma, flavor, and color [4]. During the roasting process, the coffee bean increases in size, and the silverskin, a thin tegument that covers the green coffee beans, is detached. This thin layer removed from the bean is the only co-product of the roasting stage and represents approximately 4–5% of the bean weight [8]. Due to the high volume of coffee roasted worldwide, large quantities of this co-product are generated. It is thus important to properly manage the accumulation of this co-product to reduce its environmental impact [9]. The main uses given to this co-product have been as biofuel, compost production, and soil fertilization. However, in recent years, several studies have been carried out to find new applications for coffee silverskin, due to the high content of bioactive compounds found in its composition, including dietary fiber, proteins, sugars, methylxanthines like caffeine, phenolic acids (mainly chlorogenic and neochlorogenic acids), and flavonoids, among others [1,10,11]. Due to this composition and the low fat and carbohydrate content, it would be interesting and useful to use the coffee silverskin as an innovative and functional food ingredient. In this sense, several studies explored the possibility of using coffee silverskin as an ingredient in various matrices. Cantele et al. [12] proposed the production of biscuits enriched with coffee silverskin obtained from *C. arabica* and *C. canephora*, improving the polyphenolic compound and dietary fiber content. Similarly, Dauber et al. [1] developed cookies enriched with coffee silverskin powder, obtaining a product with higher antioxidant activity and phenolic profile than the control. Bertolino et al. [13] developed a whole cow milk yogurt with added coffee silverskin and stated that silverskin can be used in yogurt production to increase the nutraceutical value of the products. Therefore, the objective of this study was to evaluate the chemical, physico-chemical, techno-functional, antioxidant, and bioactive composition, including the determination of free and bound polyphenolic compounds as well as caffeine content. Coffee silverskin derived from *Coffea arabica* and *Coffea canephora* from six different geographical origins and roasted under the same conditions was analyzed in order to assess its suitability as an ingredient for the development of novel functional food products. Unlike earlier studies that focused on single origins, mixed by-products, extract-based analyses, or specific compositional aspects, this work systematically compares coffee silverskin derived from *C. arabica* and *C. canephora* from six different geographical origins, all roasted under identical conditions. A major novelty lies in the simultaneous determination of free and bound polyphenolic compounds, offering a more complete evaluation of the antioxidant potential of coffee silverskin within the whole matrix. Additionally, caffeine content and techno-functional properties relevant to food formulation are jointly assessed, providing new insights into the influence of species and origin on the suitability of coffee silverskin for the development of novel functional food products.

## 2. Materials and Methods

### 2.1. Plant Material

The silverskin samples were obtained from two different species, *Coffea arabica* (Arabica) and *Coffea canephora* (Robusta), and supplied by the CoffeeShop company (Elche, Spain). Within these species, a distinction was made between the silverskins obtained from roasting coffee beans from six different countries: Kenya (SSFK), Guatemala (SSFG), Ethiopia (SSFE), Rwanda (SSFR), and Burundi (SSFB), belonging to the species *C. arabica*, and India (SSFI), belonging to the species *C. canephora*. The different coffee beans were subjected to a light roasting process at a temperature of 195 °C for 12 min.

To obtain the flour, the silverskins were ground in a Black + Decker coffee grinder ( Black + Decker, Towson, MD, USA) for 40 s. After that, the samples obtained were passed through a mesh (0.417 mm) and stored in vacuum bags until use.

### 2.2. Chemical Composition

The chemical composition (fat, protein, total dietary fiber, soluble dietary fiber, and insoluble dietary fiber) was determined following the official methods described by the Association of Official Agricultural Chemists [14]. The mineral content was assessed using inductively coupled plasma-mass spectrometry (ICP-MS, Shimadzu MS-2030, Shimadzu, Kioto, Japan). The ICP-MS was operated under the following conditions: carrier gas 42 L/h; plasma gas 540 L/h; auxiliary gas 66 L/h; radio frequency 1200 W; and energy filter 7.0 V. The results were expressed as mg/100 g of flour.

### 2.3. Physico-Chemical Properties

The pH of the samples was measured with a pH-meter Crison-507 (Crison, Barcelona, Spain) from the solution formed by mixing the flour with distilled water in a ratio of 1:10. The pH meter was calibrated daily using standard buffer solutions (pH 4.00, 7.00, and 10.00) at room temperature before each measurement. Automatic temperature compensation was applied during all readings. The water activity (Aw) of dry flours was directly determined at 25 °C with an electric hygrometer Novasina TH-500 (Novasina, Pfaeffikon, Switzerland). As regards color coordinates, L* (lightness), a* (redness), and b* (yellowness) of silverskin flours were evaluated. For this purpose, the different silverskin flours were placed in the granular materials attachment, compacted, and measured using a Minolta CM-700 spectrocolorimeter (Minolta Camera Co., Osaka, Japan) with the following settings: illuminant D65, SCI mode, and 10° standard observer angle.

### 2.4. Techno-Functional Properties

Water holding capacity (WHC) and oil holding capacity (OHC) were determined following the methodology described by Chau et al. [15]. The WHC of each sample was expressed as the weight of water held by 1 g of the corresponding sample, whilst OHC was expressed as the weight of oil held by 1 g of the corresponding sample. The swelling capacity (SWC) was determined following the methodology described by Robertson et al. [16], and the results were expressed as mL water per g of sample (mL/g).

### 2.5. Polyphenolic Profile

#### 2.5.1. Extraction Method

The extraction of polyphenolic compounds was separated into two fractions: (i) extracts containing free polyphenolic compounds and (ii) extracts containing bound polyphenolic compounds. To obtain the extracts with free polyphenolic compounds, the procedure described by Genskowsky et al. [17] was used. To obtain the extract with bound polyphenolic compounds, the pellet produced after the free polyphenolic compounds extraction was used, following the methodology described by Mpofu et al. [18]. The extracts obtained (free and bound) were maintained at −40 °C until high-performance liquid chromatography analysis.

#### 2.5.2. High Performance Liquid Chromatography Analysis

Polyphenolic profiles of free and bound extracts obtained from all samples were determined by high-performance liquid chromatography following the methodology described by Genskowsky et al. [17]. The identified compounds were quantified according to the peak area measurements, which were reported in calibration curves of the corresponding authentic standards. The values were expressed as mg/100 g of flour.

### 2.6. Caffeine Content

The caffeine content of the extracts obtained in Section 2.5.1 was assessed. The HPLC assay was performed with the method proposed by Grillo et al. [19]. The caffeine content was determined based on peak area measurements, which were compared to calibration curves from the authentic standard.

### 2.7. Antioxidant Analysis

#### 2.7.1. Extraction Method

Silverskin samples (1 g) were mixed with 10 mL of methanol: water (80:20, *v*/*v*) and then sonicated for 15 min. After centrifugation for 18 min, 4800× *g* at 4 °C, the supernatants were collected, and the pellets were mixed with 10 mL of acetone: water (70:30, *v*/*v*), and the same procedure was performed. Then, both supernatants were combined and evaporated to dryness in a rotary-evaporator Labtech EV311 Plus (Labtech, Sorisole BG, Italy). Eight milliliters of methanol were added to the residue, and the mixture was thoroughly shaken in a Vortex for 2 min. The methanolic extract was filtered through a 0.45 μm filter and stored at −40 °C until use.

#### 2.7.2. Methodology

The antioxidant capacity was assessed by means of the following four in vitro spectrophotometric assays. The DPPH assay was performed using the stable radical 2,2-diphenyl-1-picrylhydrazyl, following the method proposed by Brand-Williams et al. [20]. The ferric reducing antioxidant power (FRAP) was assessed by means of the potassium ferricyanide-ferric chloride method described by Oyaizu [21]. The TEAC-ABTS assay was applied following the method proposed by Gullón et al. [22]. In all three methods, Trolox was used as a reference standard, and results were expressed as mg Trolox equivalents/g of sample. Finally, the ferrous ions chelating activity (FIC) was determined by means of the method described by Mahdavi [23]. EDTA was used as a reference standard, and the results were expressed as mg EDTA/g of sample.

### 2.8. Statistical Analysis

Results obtained from chemical composition, physico-chemical and techno-functional properties, as well as antioxidant capacity of the different silverskin flours were reported as average ± standard deviation. These data, from the three independent trials, were subjected to one-way analysis of variance (ANOVA). To evaluate the statistical significance (*p* < 0.05) between the different samples. The means comparisons were made using the Tukey HDS test. The Statgraphics Centurion XVI program (Statgraphics Technologies, Inc., The Plains, VA, USA) was used for these statistical analyses.

## 3. Results and Discussions

### 3.1. Chemical Composition

Table 1 shows the chemical composition values of the different coffee silverskin flours. Regarding moisture content, the values obtained ranged between 4.50 and 6.41 g/100 g of sample, with statistical differences (*p* < 0.05) between samples. In previous studies conducted by Costa Vimercati et al. [24], the moisture content of coffee silverskin obtained from *C. arabica* was 5.03 g/100 g; more recently, Gottstein et al. [10] reported that the moisture content of silverskin obtained from *C. arabica* and *C. canephora* was 6.15 and 6.57 g/100 g, respectively. These values were consistent with those obtained in this study. Moisture is a very important factor to consider since higher moisture can be indicative of lower stability and greater susceptibility to microbiological degradation, which could affect the storage and potential use of silverskin as an ingredient in the food industry. For the protein content, the silverskin flour obtained from *C. canephora* had the highest value (*p* < 0.05). On the other hand, the flours obtained from *C. arabica* showed values contained between 12.37 and 14.21 g/100 g, with statistical differences (*p* < 0.05) among them. The values obtained in this study were lower than those reported in the scientific literature, which ranged between 17.0 and 20.0 g/100 g of sample [10,25]. The fat content of the silverskin assessed in this work ranged from 1.20 to 2.10 g/100 g. These values were slightly lower than those described by Bobková et al. [26] or Bertolino et al. [13], who reported values ranging from 2.4 to 3.5 g/100 g in coffee silverskins obtained from *C. arabica* and *C. canephora*. It is important to highlight that the samples obtained from *C. arabica* showed a higher fat content (*p* < 0.05) than those obtained for *C. canephora*. In reference to ash content, the values obtained for all samples analyzed were comprised between 4.23 and 5.34 g/100 g with statistical differences between samples (*p* < 0.05). The values obtained for this parameter, in all samples, were lower than those reported in the literature, which were around 7–8 g/100 g [27,28].

The chemical composition is strongly influenced by several factors, such as soil characteristics and agricultural practices, as well as the roasting conditions. Additionally, genetic background, cultivation environment, and processing steps collectively shape silverskin chemical composition [10].

The total dietary fiber (TDF), insoluble dietary fiber (IDF), and soluble dietary fiber (SDF) contents of the silverskin flours are summarized in Table 1. In reference to TDF content, the silverskin flours analyzed in this study showed values ranging from 71.81 to 76.86 g/100 g, with the samples cultivated in Ethiopia and Guatemala those that showed the highest value (*p* < 0.05). The total dietary fiber content of silverskin shows great variability, although it is usually between 60 and 85%. Thus, Thangavelu et al. [29] and Franca et al. [30] reported that coffee silverskin from *C. arabica* species showed a TDF content of 69.89 and 68.30 g 100 g, respectively, while Gottstein et al. [10] informed that the TDF content of *C. arabica* and *C. canephora* silverskins were 67.0 and 62.0 g/100 g values lower than those obtained in this work. However, several authors [31,32] reported similar values (around 70–75 g/100 g) to those obtained in this study. In all cases, the IDF fraction was higher than the SDF, with the sample obtained from Rwanda showing the highest (*p* < 0.05) values, without statistical differences with the flours obtained from India and Guatemala (*p* > 0.05). A high content of IDF in powders or flours obtained from co-products may be considered a benefit because IDF can be used by the food industry to increase the water-holding and oil-holding properties, as well as the swelling capacity [33]. Additionally, a high IDF content might have valuable health effects related to increasing satiety, the fecal bulk, and accelerating intestinal transit, thus promoting improved functioning of the digestive system [34].

In reference to mineral content, Table 2 displays the mineral profile of silverskin flours obtained from roasting coffee beans from six different origins. In SSFI (*C. canephora*), the predominant element (*p* < 0.05) was potassium, followed by calcium and phosphorus. On the other hand, in samples obtained from *C. arabica*, the main compound (*p* < 0.05) was calcium (except in SSFR), followed by potassium, and depending on the type of sample, in some cases phosphorus (SSFG, SSFE, and SSFR) and in others magnesium (SSFK and SSFB). Iron and sodium were also detected in considerable amounts (Table 2). The elements zinc, manganese, and copper were detected, in trace amounts, in all coffee silverskin flour analyzed.

In the scientific literature, there is a great diversity of results relating to the composition and concentration of minerals in coffee silverskin. Thus, Iriondo-DeHond et al. [35] reported that the coffee silverskin from *C. arabica* species had a micronutrient profile composed of potassium (56.0 mg/g), magnesium (5.30 mg/g), sodium (4.60 mg/g), and calcium (3.50 mg/g). Similarly, Costa et al. [25] stated that the mineral composition of coffee silverskin consisted mainly of potassium (50 mg/g), magnesium (20 mg/g), and calcium (5 mg/g). In a more recent study, Martuscelli et al. [36] reported that the predominant elements in coffee silverskin obtained from a blend of *C. arabica* and *C. canephora* were calcium (5.46 mg/g), followed by magnesium (2.22 mg/g) and phosphorus (1.46 mg/g). These differences may be attributed to the conditions in which the coffee beans were cultivated. Although all the flours correspond to the same variety, *C. arabica* (except for SSFI, which corresponds to the species *C. canephora*), the coffee beans are grown in different parts of the world, meaning that climate, soil, water, and altitude, among other factors, may influence the mineral profile of the coffee bean and, consequently, the silverskin [37,38].

### 3.2. Physico-Chemical Properties

Table 3 shows the results of the physico-chemical properties of the flours obtained from coffee silverskin of different origins. In reference to pH, the values ranged between 4.30 (SSFE) and 5.79 (SSFI), with statistical differences (*p* < 0.05) between all samples analyzed. It is important to highlight that the flour obtained from variety *C. canephora* had the highest pH value. The values obtained in this work were lower than those reported in scientific literature for coffee silverskin from different varieties and origins. Thus, Vargas-Sanchez et al. [39] reported that pH values of coffee silverskin flakes and powder from dark *C. arabica* were 6.09 and 6.30, respectively. The water activity values (Aw) varied from 0.400 to 0.586. Thus, SSFG and SSFE showed the highest values (*p* < 0.05) without statistical differences between them (*p* > 0.05). These results agree with those reported by Martuscelli et al. [36] in silverskin flour obtained from a blend of *C. arabica* and *C. canephora* varieties. The low value obtained on coffee silverskin flours examined indicates that the risk of deterioration provoked by bacterial or mold strains, enzymes, and non-enzymatic reactions is minimal [40]. With the obtained values of water activity (less than 0.586) and pH (less than 4.5 in many cases), these flours would be stable against degradation caused by the growth of microorganisms.

Color is one of the most important parameters to characterize when using flours or powders from the valorization of co-products as ingredients in the development of new foods, since these flours or powders could change the appearance of the product to which they are added. The color properties of coffee silverskin flours obtained from different origins are given in Table 3. For lightness (L*), no statistical (*p* > 0.05) differences were found among samples SSFB, SSFK, SSFE, and SSFI, with values ranging between 53.92 and 55.45, whilst SSFR and SSFG showed lower (*p* < 0.05) values without differences between them (*p* > 0.05). The L* of the flour will depend on several factors, such as the moisture content, the type of drying, and particle size. For all samples, the values obtained were higher than those reported by Quagliata et al. [41] in coffee silverskins obtained from Arabica–Robusta blends subjected to sugar-glazing treatment (L* = 38.30) and without treatment (L* = 46.82). As regards redness (a*), SSFB, SSFK, and SSFR had the highest (*p* < 0.05) values, followed by SSFB and SSFE without differences between them (*p* > 0.05). It is important to highlight that the flour obtained from *C. arabica* showed the lowest (*p* < 0.05) values for this coordinate. The values obtained were similar to those reported by Vargas-Sánchez et al. [39] on coffee silverskin powder from dark *C. arabica*. In reference to yellowness (b*), the values for this coordinate range between 21.34 and 27.20, with the SSFK sample showing the highest (*p* < 0.05) value. Again, the flour obtained from *C. arabica* showed the lowest (*p* < 0.05) values. The results of this study regarding the b* color coordinate suggest a trend towards yellow that could be related to the quality and drying process of the samples.

It is worth noting that color primarily depends on the interaction of the constitutive polyphenols and anthocyanins involved in oxidation and polymerization reactions during the coffee roasting stage. Furthermore, roasting processes lead to the degradation of proteins, and Maillard reactions occur, so melanoidins also contribute to the characteristic color of coffee silverskin [42].

### 3.3. Techno-Functional Properties

When assessing the feasibility of flours or powders derived from fruit or vegetable co-products as potential functional ingredients, one of the most critical parameters to consider is their hydration properties, including water-holding capacity, oil-holding capacity, and swelling capacity [43]. These characteristics play a key role in determining food stability and directly influence texture-related attributes such as hardness and juiciness [44]. Figure 1 shows the results of the techno-functional properties of the different flours obtained from coffee silverskin. Regarding WHC, SSFB and SSFR had the highest (*p* < 0.05) WHC of all coffee silverskin flours analyzed, with values of 5.46 and 5.10 g water/g flour, respectively. On the other hand, SSFG showed the lowest (*p* < 0.05) value (3.76 g water/g flour). In this case, the species *C. arabica* or *C. canephora* did not influence this property.

The results obtained were similar to those reported by Ballesteros et al. [45], who reported that the powder obtained from mixtures of *C. arabica* or *C. canephora* coffee species had a WHC of 5.11 g water/g sample. On the other hand, Thangavelu et al. [29] reported that the WHC of coffee silverskin from arabica species treated with ultrasound ranged between 3.03 and 3.62 g water/g sample. Vargas-Sanchez et al. [39] reported that the WHC value of coffee silverskin powder from dark *C. arabica* was 2.9 g water/g sample. The variability in WHC values among the different samples and with the studies found in the literature may be due to several factors, such as granulometry of flours, porosity, insoluble dietary fiber content, and hydrophilic surface group availability that generally influence the WHC [46].

In reference to the oil holding capacity, SSFE, SSFR, SSFK, and SSFB had an OHC that ranged between 3.41 and 3.80 g oil/g sample, with no statistically significant differences (*p* > 0.05) between samples. In this case, the sample SSFI, which belongs to the species *canephora*, showed the lowest value (*p* < 0.05) for this property. The values obtained in this work were similar to those reported by Martuscelli et al. [36] in silverskin flour obtained from a blend of *C. arabica* and *C. canephora* species, who stated that this flour had an OHC of 3.02 g oil/g sample. However, Ateş and Elmacı [47] reported that coffee silverskins from the *arabica* species showed an OHC of 4.80 g oil/g sample, higher than reported in this study. OHC depends on several factors, including the chemical structure of the matrix, viscosity, total charge density, as well as the thickness and hydrophobic nature of the particle, as mentioned by Wang et al. [48] and Song et al. [49].

Finally, for swelling capacity, no statistical differences (*p* > 0.05) were found between the samples belonging to *C. arabica* (SSFG, SSFE, SSFR, SSFK, and SSFB) with values ranging between 4.48 and 5.45 mL/g sample. For this parameter (SWC), the flour obtained from the silverskin of *C. canephora* beans (SSFI) had the lowest value (4.34 mL/g sample). To the best of our knowledge, no studies have been reported in the scientific literature that have analyzed the swelling capacity of flours or powders derived from coffee silverskin. Swelling capacity is influenced by factors such as chemical structure (with higher soluble fiber content resulting in increased swelling capacity), porosity, and particle size [50]. A high swelling capacity is particularly advantageous in food applications, as it enhances product texture and overall palatability [51]. For instance, flours or powders that exhibit substantial swelling capacity can improve the juiciness of meat formulations, the structural consistency of baked goods, and the functional properties of functional food products [52].

### 3.4. Polyphenolic Profile

The polyphenolic profile of silverskin flours obtained from roasted coffee beans from six distinct geographical origins reveals substantial variability in both the qualitative and quantitative distribution of free and bound phenolic compounds, as shown in Table 4. It is important to highlight that, for all silverskin flour samples analyzed, the concentration of free polyphenolic compounds was lower than that of bound polyphenolic compounds. In the free fraction, seven compounds were identified in all samples, while in the bound fraction 5 compounds were detected. Statistically significant differences (*p* < 0.05) demonstrate that origin exerts a decisive influence on both free and bound phenolic composition, suggesting that environmental factors, post-harvest processing, and roasting behavior collectively shape the bioactive content of coffee silverskin. Overall, the results highlight the complexity of the coffee co-product matrix and underscore its potential as a functional ingredient with origin-dependent bioactive properties.

In the free polyphenolic compounds fraction, marked disparities (*p* < 0.05) emerge among the samples. For instance, free vanillin concentrations vary significantly, ranging from 0.41 µg/g in SSFG to 1.63 µg/g in SSFE, differing significantly from all other origins (*p* < 0.05). A similar pattern is observed for free ferulic acid, which is most abundant (*p* < 0.05) in SSFE (2.60 µg/g) and SSFB (2.55 µg/g). The quinic acid derivatives display the most striking variation: SSFR and SSFK exhibit high levels of 3-caffeoylquinic acid (47.44 and 29.79 µg/g, respectively), whereas SSFG shows no detectable levels. These large inter-origin discrepancies, supported by significant statistical differences (*p* < 0.05) between samples, indicate that the stability of 3-caffeoylquinic acid during roasting may vary by bean type or roasting associated with the original raw material. Similarly, 4-caffeoylquinic, 5-caffeoylquinic, and 4,5-dicaffeoylquinic acids display heterogeneous distributions. The total free polyphenols show a distinct pattern: SSFR exhibits the highest (*p* < 0. 05) content (75.65 µg/g), followed by SSFK (64.95 µg/g), whereas SSFG had the lowest (*p* < 0. 05) concentration. On the other hand, the bound polyphenolic fraction provides even more remarkable contrasts, especially for compounds such as caffeic acid, ferulic acid, and vanillic acid derivatives. The bound caffeic acid content ranges from only 13.84 µg/g in SSFG to an outstanding 65.11 µg/g in SSFE, with significant differences (*p* < 0. 05) between all origins. Bound ferulic acid shows the highest concentration in SSFI, with 109.10 µg/g, almost double or more compared with the other samples. One fact worth highlighting is that the silverskin flours obtained from *C. canephora* have a higher content of these bioactive compounds than the silverskin flours obtained from *C. arabica*. These results agree with Mannino et al. [53] and Panusa et al. [54], who reported that the polyphenolic content of *C. arabica* beans and their by-products is lower than found in *C. canephora*.

The observation that bound polyphenols predominate over free polyphenols in all samples is consistent with reports for numerous plant matrices and agro-food by-products. Several studies have shown that a substantial fraction of phenolic compounds is esterified or non-covalently associated with cell wall components, such as structural polysaccharides and lignin, which limits their extractability using conventional solvents [55,56]. This matrix–phenol association explains why free polyphenols usually represent only a minor fraction of the total phenolic content. In this context, the predominance of bound polyphenols suggests a strong interaction with the plant matrix, characteristic of fiber-rich tissues, and underscores the need to consider this fraction for a more realistic assessment of the functional potential of the ingredient [57].

Roasting can markedly affect the balance between free and bound polyphenols. Thermal treatment may partially cleave bonds between phenols and the cell wall, promoting the release of previously bound compounds, while higher roasting intensities can induce thermal degradation or chemical transformation of certain polyphenols [58]. Consequently, the net effect of roasting depends on the degree and duration of the treatment: light to medium roasting may enhance the release of bound phenols, whereas more intense roasting can reduce total phenolic content. These dynamics provide a mechanistic explanation for observed shifts in the proportion of free and bound polyphenols after roasting and should be taken into account when interpreting compositional data [59]. From a nutritional and functional perspective, the high proportion of bound polyphenols carries important implications for bioaccessibility and bioactivity. While free polyphenols have traditionally been considered the primary bioactive fraction, increasing evidence shows that bound polyphenols can be released during gastrointestinal digestion, particularly in the colon via microbial action, contributing to antioxidant and metabolic effects at the intestinal level [60,61]. This delayed release can extend the functional impact of the ingredient beyond the immediately bioaccessible fraction. Therefore, the predominance of bound polyphenols should not be viewed as a limitation, but rather as a feature with potential physiological relevance, especially in fiber-rich foods and plant co-products.

The content and concentration of polyphenolic compounds in silverskin depend on several factors, such as coffee species and variety, environmental conditions of cultivation, origin, and fundamentally on the conditions of time and temperature in which roasting takes place [27,62]. In this context, Nzekoue et al. [8] reported that in *C. arabica* silverskin obtained from beans of Ethiopian origin, eighteen phenolic compounds with 3-caffeoylquinic acid, 5-caffeoylquinic acid, and 3,5-dicaffeoylquinic acid were detected as the most abundant polyphenols, with values compressed between 3.11 mg/g and 5.44 mg/g. More recently, Giordano et al. [63] reported that the principal polyphenolic compounds found in silverskin obtained from *C. arabica* and *C. canephora* dark roasted beans were 5-caffeoylquinic acid with values of 0.61 and 0.48 mg/g, respectively, and dicaffeoylquinic acid with values of 0.14 and 0.30 mg/g for *C. arabica* and *C. canephora* silverskin. Similarly, Peixoto et al. [64] stated that the main polyphenolic compounds found in coffee silverskin obtained from a blend of both Arabica and Robusta were 4-feruloylquinic acid, 5-caffeoylquinic acid, and 5-feruloylquinic acid with values of 0.39, 1.04, and 1.05 mg/g, respectively. Several recent studies have investigated how phenolic compounds such as caffeoylquinic acids and derivatives such as ferulic acid are related to the sensory characteristics of coffee and other food products. In this sense, Linne et al. [65] reported that 3-caffeoylquinic acid and 4-caffeoylquinic acid isomers can influence mouthfeel attributes such as “body” and oral coating in coffee, with perceptible sensory effects even at ecologically relevant concentrations, although not in a linear manner with respect to compound concentration. In a similar study, Tieghi et al. [66] reported that in specialty coffees, the presence of phenolic acids, including ferulic acid, was statistically correlated with sensory scores such as sweetness and acidity, which suggests a clear relationship between chemical profile and perceived quality. In addition, research evaluating coffee blends with differing contents of caffeoylquinic acids has reported variations in attributes such as astringency and overall sensory appeal, further supporting the relevance of these compounds to coffee sensory perception [67].

### 3.5. Caffeine Content

Caffeine, a methylxanthine, is another main type of phytochemical present in coffee silverskin. This phytocompound has been related to several positive effects on human health, acting as anticarcinogenic, anti-obesity, or diuretic agents, among other effects [68]. Figure 2 shows the caffeine content of coffee silverskin flours of different origins. SSFB showed the highest (*p* < 0.05) value (7.32 mg/g), followed by SSFK, SSFE, SSFI, and SSFR with values of 5.83, 5.35, 4.37, and 4.23 mg/g, respectively, without statistical differences (*p* > 0.05) between them. Finally, SSFG showed the lowest (*p* < 0.05) value (0.85 mg/g). In the scientific literature, there is significant variation in the concentration of caffeine present in silverskin co-products, since this concentration depends on many factors such as the coffee variety, growing conditions, the origin, the roasting process, the extraction methods, and silverskin storage and collection methods, among others [8]. Thus, the values obtained in this work were similar to those reported by Machado et al. [62], who reported a caffeine content of silverskin obtained from a blend of *C. arabica* and *C. canephora*, approximately 40%:60%, respectively, of 7.1 mg/g. Gottstein et al. [10] reported that the caffeine content of *C. arabica* and *C. canephora* silverskins was 8.0 and 8.6 mg/g. Similarly, Bessada et al. [27] reported that the caffeine content in coffee silverskin from five different geographical origins ranged between 7.1 and 12.15 mg/g. On the other hand, Quagliata et al. [41] reported that silverskin obtained from Arabica–Robusta blends subjected to sugar-glazing had a caffeine content of 13.2 mg/g, while the silverskin obtained from 100% Arabica without the torrefacto treatment showed a value of 8.70 mg/g. The recovery of caffeine from coffee silverskin deeply depends on the extraction methodology. Thus, the use of novel technologies, such as ultrasonic-assisted extraction (UAE) or Pressurized Liquid Extraction (PLE) process, substantially increased the concentration of this compound. In this sense, Peixoto et al. [64] reported that the caffeine content of coffee silverskin obtained from a blend of both *C. arabica* and *C. canephora* beans and extracted using UAE was 27.70 mg/g. Similarly, Koskinakis et al. [69] reported a value of 56.7 mg/g dry extract from coffee silverskin extracted by means of the PLE process.

### 3.6. Antioxidant Capacity

To thoroughly evaluate the antioxidant activity of a flour or powder derived from the valorization of co-products, it is necessary to employ more than one analytical methodology that encompasses different mechanisms of action. In this study, four distinct assays were utilized: ABTS, FRAP, FIC, and DPPH. Figure 3A–D illustrates the results obtained for the antioxidant capacity measurement of silverskin flours obtained from roasting coffee beans from six different origins. For the DPPH assay (Figure 3A), no significant differences (*p* > 0.05) were observed among SSFI, SSFG, SSFK, and SSFB, while SSFR and SSFE showed lower (*p* < 0.05) values without statistical differences (*p* > 0.05) between them. In the FRAP assay (Figure 3B), which measures the ability to reduce ferric ions, the analyzed silverskin flours showed a wide range of antioxidant capacities, with values ranging between 9.69 (SSFE) and 19.68 mg Trolox equivalents (TE)/g of sample (SSFK) with statistically significant differences (*p* < 0.05) between samples. For the ABTS assay (Figure 3C), the SSFE sample showed the highest (*p* < 0.05) value (11.05 mg TE/g), followed by SSFK (10.02 mg TE/g), whilst SSFR and SSFI showed the lowest (*p* < 0.05) values without statistical differences (*p* > 0.05) between them. Finally, for the ferrous ion chelating activity (Figure 3D), the results indicated that SSFI had the highest (*p* < 0.05) chelating capacity without statistical differences (*p* > 0.05) with SSFE.

In the scientific literature, there is a lot of variability regarding the results obtained for the antioxidant activity of extracts or flours derived from coffee silverskin. This variability is attributable to multiple factors, including the species of coffee used, the method employed, and the type of extraction, among others. Furthermore, the different ways in which the results obtained can be expressed make comparing data extremely difficult. In this context, Del Castillo et al. [70] indicates that the silverskin obtained from *C. canephora* beans exhibits higher antioxidant activity, as measured by DPPH and ABTS assays, with values of 23.1 and 22.5 mmol TE/100 g, respectively, compared to samples obtained silverskin from *C. arabica* beans, which yielded values of 22.5 and 8.5 mmol TE/100 g. More recently, Nzekoue et al. [8] reported that with the DPPH assay, the methanolic extract obtained from coffee (*C. arabica* beans of Ethiopian origin) silverskin had an IC_50_ (which is the concentration of the extract necessary to cause 50% of DPPH inhibition) of 101.7 µg/mL. In a study carried out by Vargas-Sanchez et al. [39], the antioxidant capacity, measured with ABTS and FRAP assay, of coffee silverskin powder from dark *C. arabica* was analyzed. These authors found that this extract had % ABTS^·+^ inhibition of 87.89%, whereas the FRAP value was 2.65 mg Fe^2+^g sample. Franca et al. [30] stated that the antioxidant capacity of methanolic and ethanolic coffee silverskin extracts measured with the FRAP assay was 170.64 and 98.37 µmol Fe_2_SO_4_/g, respectively, while the DPPH-IC_50_ for methanolic and ethanolic coffee silverskin extracts was 1727.6 and 251.15 (g/g DPPH). Dauber et al. [1] reported that the antioxidant capacity of coffee silverskin obtained from a blend of *C. arabica* and *C. canephora*, and measured with ABTS assay, was 49.8 μmol TE/g sample. The antioxidant activity of coffee silverskin might be due to its rich profile of polyphenolic compounds, particularly chlorogenic acids and related caffeoylquinic derivatives. In this way, Dong et al. [71] showed that fractions obtained from coffee silverskin, which contain 3-caffeoylquinic acid, feruloylquinic acids, and various dicaffeoylquinic acids, exhibit the highest antioxidant capacity. These compounds, well known for their strong radical-scavenging ability, were especially concentrated in the free phenolic fraction, which correlated directly with superior in vitro antioxidant performance. Complementing these findings, Jirarat et al. [72] demonstrated that extraction techniques optimized for phenolic release significantly enhance the antioxidant potential of silverskin extracts. Their results confirmed that chlorogenic acids, other phenolic acids, and additional polyphenols are the dominant contributors to antioxidant activity, while caffeine and interactions with dietary fiber provide secondary roles.

## 4. Conclusions

The results obtained highlight the considerable potential of coffee silverskin, generated during the roasting process, as a promising ingredient for the development of new food products. This potential is primarily attributed to (i) its high dietary fiber content, and (ii) its notable concentration of bioactive compounds of interest, such as caffeine and polyphenolic compounds. From a circular economy perspective, the valorization of coffee silverskin as a food ingredient represents a high-value upcycling strategy that closes material loops within the coffee production chain. Given that silverskin is the only solid co-product generated during roasting and accounts for approximately 4–5% of bean weight, its conversion into a functional ingredient can significantly reduce waste streams while generating added economic value. Moreover, the demonstrated techno-functional properties allow its incorporation without extensive chemical modification, aligning with clean-label trends and sustainable food design.

Thus, silverskin flour from India (*C. canephora*) stands out for having the highest protein content, highest pH, and remarkably high bound polyphenol content, particularly ferulic and caffeic acids, together with strong metal chelating activity (FIC). This profile suggests its suitability for nutritionally enriched formulations such as fiber-rich supplements or functional snacks. Silverskin flour from Guatemala (*C. arabica*) exhibits the highest total dietary fiber content, with a particularly high soluble dietary fiber fraction, but the lowest polyphenol and caffeine contents. This makes it especially suitable for fiber enrichment purposes in food products where low stimulant content and mild bioactivity are desirable, such as bakery products or fiber-enhanced foods. Silverskin flour from Ethiopia (*C. arabica*) shows a balanced composition, combining high total dietary fiber, elevated bound polyphenol content, high oil-holding capacity, and strong antioxidant activity (ABTS). This sample appears well-suited for functional foods requiring lipid retention and antioxidant protection, such as meat analogues or emulsified products. Silverskin flour from Rwanda (*C. arabica*) is characterized by the highest insoluble dietary fiber content, very high free polyphenol concentration, particularly caffeoylquinic acids, and excellent water-holding capacity. These properties make it particularly attractive for applications aimed at texture enhancement, satiety promotion, and antioxidant delivery, such as baked goods or high-fiber formulations. Silverskin flour from Kenya (*C. arabica*) combines high antioxidant capacity (FRAP and ABTS), high caffeine content, good swelling capacity, and low water activity, indicating strong stability. This profile supports its potential use in energy-oriented or stimulant functional foods and formulations requiring high oxidative stability. Finally, Silverskin flour from Burundi (*C. arabica*) exhibits high water- and oil-holding capacities, high soluble dietary fiber, high caffeine content, and a rich bound polyphenolic profile. These characteristics suggest its suitability for food formulations demanding moisture retention and enhanced mouthfeel, such as bakery, dairy, or plant-based products.

It is also important to consider that, prior to its application as a food ingredient, the possible formation of roasting-derived compounds, such as 5-hydroxymethylfurfural or acrylamide, must be assessed, as these substances may exert detrimental effects both on the food matrix to which the ingredient is added and on the health of the final consumer.

## Figures and Tables

**Figure 1 foods-15-00097-f001:**
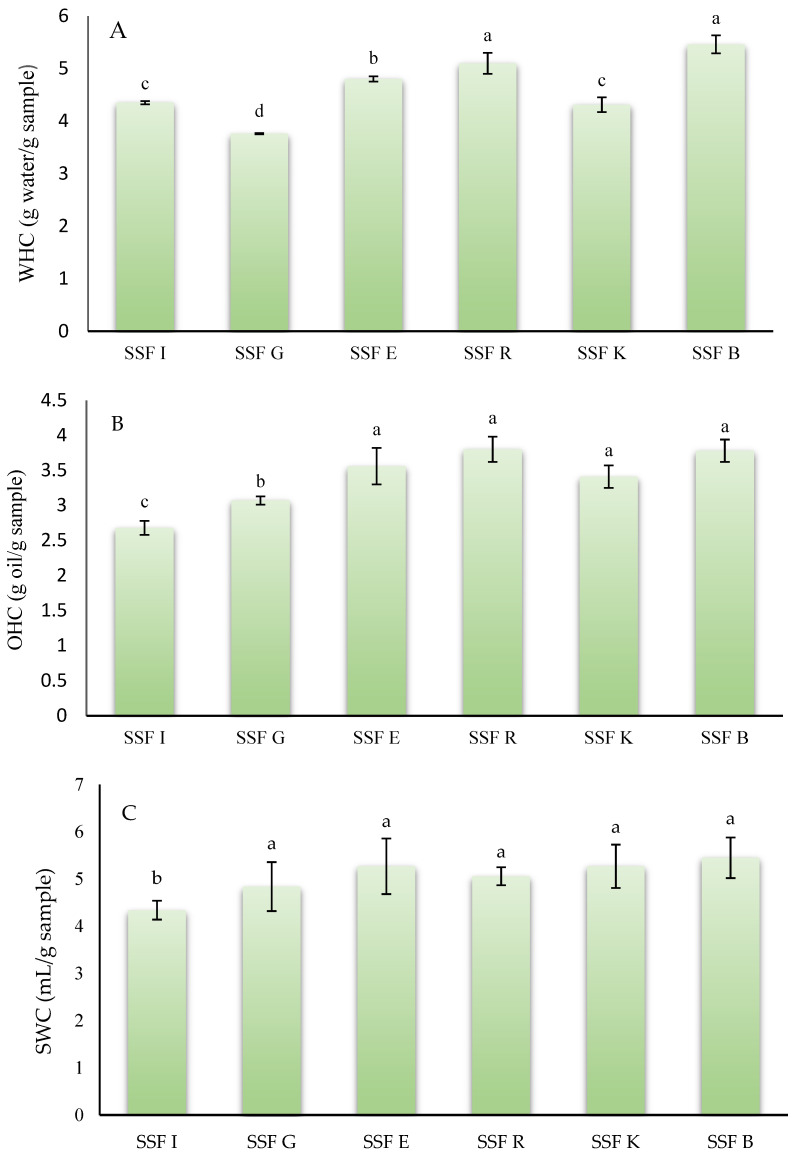
Techno-functional properties of silverskin flours obtained from roasting coffee beans from six different origins. (**A**): water holding capacity; (**B**): oil holding capacity; (**C**): swelling capacity. SSFI: coffee silverskin flour from India; SSFG: Coffee silverskin flour from Guatemala; SSFE: coffee silverskin flour from Ethiopia; SSFR: coffee silverskin flour from Rwanda; SSFK: coffee silverskin flour from Kenia; SSFB: Coffee silverskin flour from Burundi. For the same analysis (WHC, OHC, and SWC), histograms with different letters indicate significant differences (*p* < 0.05) according to Tukey’s multiple range test.

**Figure 2 foods-15-00097-f002:**
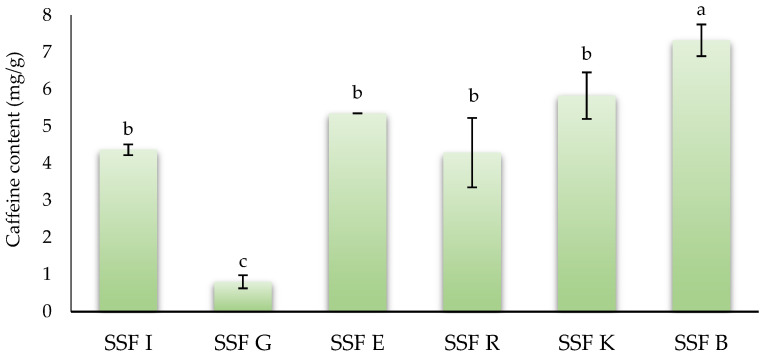
Caffeine content of silverskin flours obtained from roasting coffee beans from six different origins. SSFI: coffee silverskin flour from India; SSFG: coffee silverskin flour from Guatemala; SSFE: coffee silverskin flour from Ethiopia; SSFR: coffee silverskin flour from Rwanda; SSFK: coffee silverskin flour from Kenia; SSFB: coffee silverskin flour from Burundi. Histograms with different letters indicate significant differences (*p* < 0.05) according to Tukey’s multiple range test.

**Figure 3 foods-15-00097-f003:**
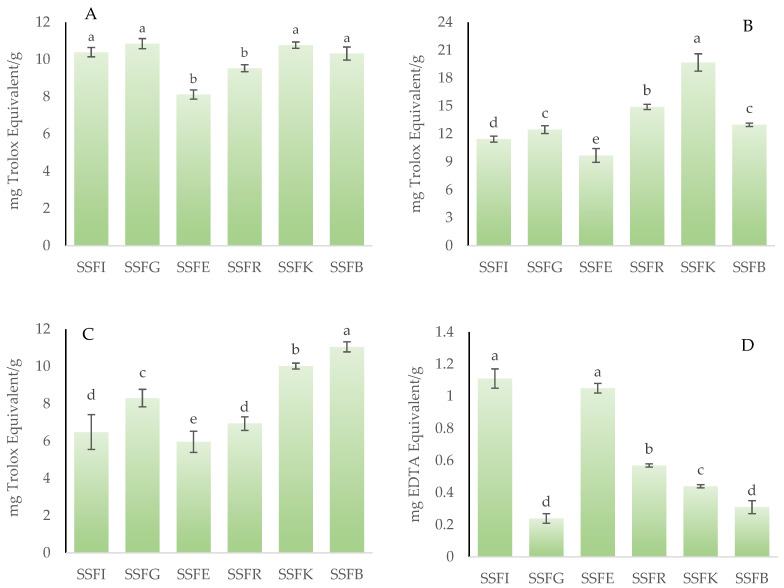
Antioxidant capacity of silverskin flours obtained from roasting coffee beans from six different origins. (**A**): measured with DPPH assay; (**B**): measured with FRAP assay; (**C**): measured with ABTS assay; (**D**): measured with FIC assay. SSFI: coffee silverskin flour from India; SSFG: coffee silverskin flour from Guatemala; SSFE: coffee silverskin flour from Ethiopia; SSFR: coffee silverskin flour from Rwanda; SSFK: coffee silverskin flour from Kenia; SSFB: coffee silverskin flour from Burundi. For each antioxidant assay, histograms with different letters indicate significant differences (*p* < 0.05) according to Tukey’s multiple range test.

**Table 1 foods-15-00097-t001:** Chemical composition of silverskin flours obtained from roasting coffee beans from six different origins.

	Moisture	Protein	Fat	Ash	TDF	IDF	SDF
SSFI	4.40 ± 0.39 ^c^	15.08 ± 0.32 ^a^	1.20 ± 0.07 ^c^	4.65 ± 0.27 ^b^	74.30 ± 0.62 ^b^	58.35 ± 1.20 ^ab^	15.95 ± 0.53 ^b^
SSFG	4.50 ± 0.15 ^c^	12.37 ± 0.10 ^d^	1.70 ± 0.08 ^b^	4.25 ± 0.52 ^bc^	76.86 ± 1.23 ^a^	58.68 ± 0.55 ^ab^	18.18 ± 0.76 ^a^
SSFE	4.94 ± 0.28 ^bc^	13.10 ± 0.03 ^cd^	1.85 ± 0.07 ^b^	4.23 ± 0.25 ^c^	75.21 ± 2.65 ^a^	55.88 ± 0.52 ^c^	19.33 ± 0.94 ^a^
SSFR	5.69 ± 0.18 ^b^	12.98 ± 0.31 ^d^	2.10 ± 0.12 ^a^	5.34 ± 0.22 ^a^	72.35 ± 0.40 ^b^	60.58 ± 0.48 ^a^	11.77 ± 0.22 ^d^
SSFK	6.41 ± 0.44 ^a^	13.94 ± 0.10 ^bc^	1.88 ± 0.08 ^b^	5.01 ± 0.20 ^ab^	71.81 ± 1.33 ^c^	57.81 ± 1.25 ^b^	14.00 ± 0.47 ^c^
SSFB	5.26 ± 0.22 ^b^	14.21 ± 0.12 ^b^	1.96 ± 0.04 ^ab^	4.91 ± 0.25 ^ab^	73.09 ± 1.24 ^b^	54.02 ± 1.58 ^c^	19.07 ± 2.19 ^a^

Values expressed as g/100 g. TDF: Total dietary fiber; IDF: insoluble dietary fiber; SDF: soluble dietary fiber; SSFI: coffee silverskin flour from India; SSFG: coffee silverskin flour from Guatemala; SSFE: coffee silverskin flour from Ethiopia; SSFR: coffee silverskin flour from Rwanda; SSFK: coffee silverskin flour from Kenia; SSFB: coffee silverskin flour from Burundi. Values with different letters within the same column indicate significant differences (*p* < 0.05) according to Tukey’s multiple range test.

**Table 2 foods-15-00097-t002:** Mineral profile of silverskin flours obtained from roasting coffee beans from six different origins.

	Calcium	Iron	Potassium	Magnesium	Sodium	Phosphorus
SSFI	9.25 ± 0.19 ^dB^	0.19 ± 0.02 ^bF^	17.59 ± 0.30 ^aA^	5.10 ± 0.04 ^aD^	0.45 ± 0.13 ^bE^	5.30 ± 0.11 ^aC^
SSFG	16.44 ± 0.18 ^aA^	0.20 ± 0.01 ^bF^	12.53 ± 0.22 ^cB^	2.25 ± 0.04 ^cD^	0.66 ± 0.10 ^aE^	2.74 ± 0.08 ^cC^
SSFE	13.31 ± 0.33 ^bA^	0.12 ± 0.01 ^dE^	12.69 ± 0.25 ^cB^	3.53 ± 0.10 ^bC^	0.66 ± 0.04 ^aD^	3.99 ± 0.06 ^bC^
SSFR	10.58 ± 0.14 ^cB^	0.26 ± 0.01 ^aF^	14.44 ± 0.63 ^bA^	2.22 ± 0.07 ^cD^	0.56 ± 0.04 ^aE^	5.23 ± 0.07 ^aC^
SSFK	12.24 ± 0.06 ^bA^	0.20 ± 0.01 ^bF^	8.68 ± 0.37 ^dB^	2.09 ± 0.06 ^cC^	0.53 ± 0.03 ^aE^	1.01 ± 0.08 ^dD^
SSFB	13.25 ± 0.07 ^bA^	0.16 ± 0.02 ^cF^	6.66 ± 0.10 ^eB^	2.57 ± 0.02 ^cC^	0.63 ± 0.06 ^aE^	1.02 ± 0.05 ^dD^

Values expressed as mg/g. SSFI: coffee silverskin flour from India; SSFG: coffee silverskin flour from Guatemala; SSFE: coffee silverskin flour from Ethiopia; SSFR: coffee silverskin flour from Rwanda; SSFK: coffee silverskin flour from Kenia; SSFB: coffee silverskin flour from Burundi. For the same flour, values with different capital letters within the same row indicate significant differences (*p* < 0.05) according to Tukey’s multiple range test. For the same compound, values with different small letters within the same column indicate significant differences (*p* < 0.05) according to Tukey’s multiple range test.

**Table 3 foods-15-00097-t003:** Physico-chemical Properties of silverskin flours obtained from roasting coffee beans from six different origins.

	pH	Aw	L*	a*	b*
SSFI	5.79 ± 0.01 ^a^	0.541 ± 0.005 ^c^	54.60 ± 0.38 ^a^	5.77 ± 0.04 ^d^	21.34 ± 0.13 ^d^
SSFG	4.47 ± 0.03 ^e^	0.586 ± 0.001 ^a^	51.41 ± 1.31 ^b^	7.95 ± 0.24 ^c^	22.18 ± 0.40 ^c^
SSFE	4.30 ± 0.01 ^f^	0.579 ± 0.001 ^a^	53.92 ± 1.22 ^a^	7.94 ± 0.18 ^c^	23.28 ± 0.51 ^b^
SSFR	5.15 ± 0.01 ^b^	0.568 ± 0.001 ^b^	51.85 ± 0.22 ^b^	8.89 ± 0.22 ^a^	22.34 ± 0.11 ^c^
SSFK	4.68 ± 0.01 ^d^	0.400 ± 0.000 ^e^	55.45 ± 1.37 ^a^	8.40 ± 0.18 ^b^	27.20 ± 0.38 ^a^
SSFB	4.96 ± 0.00 ^c^	0.420 ± 0.000 ^d^	53.99 ± 0.40 ^a^	8.96 ± 0.19 ^a^	26.50 ± 0.37 ^b^

SSFI: coffee silverskin flour from India; SSFG: coffee silverskin flour from Guatemala; SSFE: coffee silverskin flour from Ethiopia; SSFR: coffee silverskin flour from Rwanda; SSFK: coffee silverskin flour from Kenia; SSFB: Coffee silverskin flour from Burundi. Values with different letters within the same column indicate significant differences (*p* < 0.05) according to Tukey’s multiple range test.

**Table 4 foods-15-00097-t004:** Polyphenolic profile of silverskin flours obtained from roasting coffee beans from six different origins.

	SSFI	SSFG	SSFE	SSFR	SSFK	SSFB
Compound	Free					
Vanillin	1.00 ± 0.01 ^cE^	0.41 ± 0.11 ^dC^	1.63 ± 0.04 ^aD^	0.65 ± 0.08 ^dE^	1.43 ± 0.01 ^bF^	1.17 ± 0.11 ^cD^
Ferulic acid	1.61 ± 0.02 ^bD^	0.17 ± 0.00 ^cC^	2.60 ± 0.01 ^aC^	1.69 ± 0.02 ^bD^	1.68 ± 0.10 ^bF^	2.55 ± 0.40 ^aC^
3-caffeoylquinic acid	0.87 ± 0.04 ^eE^	nd	1.67 ± 0.05 ^dD^	47.44 ± 0.08 ^aA^	29.79 ± 1.80 ^bA^	4.11 ± 1.23 ^cB^
4-caffeoylquinic acid	2.07 ± 0.42 ^cC^	1.55 ± 0.02 ^cB^	1.83 ± 0.32 ^cD^	4.99 ± 0.06 ^bC^	9.21 ± 1.42 ^aC^	4.67 ± 0.83 ^bB^
5-caffeoylquinic acid	2.29 ± 0.34 ^cC^	2.13 ± 0.34 ^cA^	2.40 ± 0.45 ^cC^	8.42 ± 0.75 ^aB^	3.69 ± 0.32 ^bE^	3.78 ± 0.51 ^bC^
4,5-dicaffeoylquinic acid	7.30 ± 2.40 ^cA^	2.53 ± 0.38 ^dA^	9.98 ± 0.04 ^bA^	7.52 ± 0.54 ^cB^	12.48 ± 1.70 ^aB^	12.96 ± 1.01 ^aA^
Caffeic acid derivative	4.57 ± 0.04 ^bB^	1.55 ± 0.03 ^dB^	5.90 ± 0.08 ^aB^	4.94 ± 0.75 ^abC^	6.67 ± 1.06 ^aD^	2.91 ± 0.44 ^cC^
**Total polyphenols**	**19.71 ± 3.27 ^e^**	**8.34 ± 0.88 ^f^**	**26.01 ± 0.99 ^d^**	**75.65 ± 2.28 ^a^**	**64.95 ± 6.41 ^b^**	**32.15 ± 4.53 ^c^**
Compound	Bound					
Vanillic acid derivative	3.75 ± 1.00 ^dE^	21.70 ± 0.01 ^bA^	13.04 ± 0.01 ^cD^	26.05 ± 2.26 ^bB^	54.26 ± 3.97 ^aA^	48.39 ± 3.23 ^aB^
Caffeic acid	58.25 ± 1.25 ^bB^	13.84 ± 0.09 ^eC^	65.11 ± 0.06 ^aA^	45.92 ± 3.76 ^cA^	23.86 ± 2.49 ^dB^	47.88 ± 0.89 ^cB^
Ferulic acid	109.10 ± 0.05 ^aA^	12.22 ± 0.11 ^eD^	64.58 ± 0.03 ^bA^	14.83 ± 2.77 ^eC^	24.02 ± 2.41 ^dB^	29.03 ± 1.35 ^cC^
4-Hydroxybenzoic acid	18.28 ± 2.60 ^aD^	9.49 ± 0.07 ^bE^	16.89 ± 2.35 ^aC^	2.86 ± 0.33 ^cD^	10.35 ± 0.39 ^bC^	1.60 ± 1.33 ^cD^
4,5-dicaffeoylquinic acid	48.29 ± 0.02 ^cC^	17.39 ± 2.05 ^fB^	41.59 ± 0.01 ^dB^	30.42 ± 2.80 ^eB^	54.46 ± 1.91 ^bA^	60.58 ± 1.49 ^aA^
**Total polyphenols**	**219.39 ± 4.32 ^a^**	**65.15 ± 2.26 ^e^**	**184.32 ± 0.11 ^b^**	**117.22 ± 11.59 ^d^**	**156.60 ± 10.78 ^c^**	**185.88 ± 8.96 ^b^**

Values expressed as µg/g. nd: not detected. SSFI: coffee silverskin flour from India; SSFG: coffee silverskin flour from Guatemala; SSFE: coffee silverskin flour from Ethiopia; SSFR: coffee silverskin flour from Rwanda; SSFK: coffee silverskin flour from Kenia; SSFB: coffee silverskin flour from Burundi. For the same treatment (free and bound) values with different capital letters within the same column indicate significant differences (*p* < 0.05) according to Tukey’s multiple range test. Values with different small letters within the same row indicate significant differences (*p* < 0.05) according to Tukey’s multiple range test.

## Data Availability

The raw data supporting the conclusions of this article will be made available by the authors on request.

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
