# Peer review of "Characterization of Coffee Silverskin from Different Origins to Evaluate Its Potential as an Ingredient in Novel Food Products"

_foods, 2025, doi:10.3390/foods15010097_

Round 1

Reviewer 1 Report

Comments and Suggestions for Authors

The article written by Candela-Salvador et al. focuses on the chemical properties of coffee silverskin. The manuscript is of good quality, however I have some suggestions, which, I think, allow to improve it.

  • The novelty of the work is not explained properly. There are several works published already in this topic, e.g.:
  • Chemical, functional, and structural properties of spent coffee grounds and coffee silverskin, LF Ballesteros, JA Teixeira, SI Mussatto - Food and bioprocess technology, 2014
  • Spent coffee grounds and coffee silverskin as potential materials for packaging: A review CV Garcia, YT Kim - Journal of Polymers and the Environment, 2021
  • Chemical composition, antioxidant and enzyme inhibitory properties of different extracts obtained from spent coffee ground and coffee silverskin, G Zengin, KI Sinan, MF Mahomoodally, S Angeloni… - Foods, 2020
  • Chemical characterization and antioxidant properties of a new coffee blend with cocoa, coffee silverskin and green coffee minimally processed, VS Ribeiro, AE Leitão, JC Ramalho… - Food Research …, 2014
  • Use of coffee silverskin to improve the functional properties of cookies, D Gocmen, Y Sahan, E Yildiz, M Coskun… - Journal of food science …, 2019
  • Coffee silver skin: Chemical characterization with special consideration of dietary fiber and heat-induced contaminants, V Gottstein, M Bernhardt, E Dilger, J Keller… - Foods, 2021
  • The title is suitable, but the abstract should contain more direct results (numerical values) to support the statements about the results. E.g. the range of minerals concentrations detected.
  • Chapter 2.3. Physico-chemical properties – it is not clear if the tests are conducted for dry powder/suspension/solution. Please be more specific how the samples were prepared for the measurements.
  • The Results sections contain many tables. Maybe some of them can be replaced with graphs. Moreover, the existing graphs can be smaller and more elaborated graphically.
  • There is a lack of summary characterizing the different samples. Which one is the best for specific purposes?
  • Conclusions should be also more specific, e.g. which kind of coffee silverskin is the most rich in nutrients?

Reviewer 2 Report

Comments and Suggestions for Authors

This manuscript is rich in content, features a clear research concept, and has a comprehensive experimental design. The study systematically analyzed the composition of coffee silverskin from six different origins. The core finding—that silverskin is rich in dietary fiber (~75%) and high levels of bioactive compounds—strongly supports its significant potential as a novel functional food ingredient, aligning with current circular economy trends in the food industry. However, there is still room for improvement in the in-depth discussion of the results and crucial methodological details.

  1. Methodology Transparency: Missing Raw Material Information The manuscript does not specify the degree of roasting (e.g., light, medium, dark) of the coffee beans from which the silverskin samples were generated. The roasting level is a crucial factor determining the silverskin's final composition. Suggestion: Supplement the description of the roasting degree of the coffee beans used in Section 2.1.
  2. Authors must add 2–3 sentences in the Discussion with proper references.
  3. Depth and Consistency of Results Discussion The discussion repeatedly notes that certain values (e.g., protein, ash, pH) are lower than or differ from scientific literature.The authors should provide a more in-depth explanation for these deviations by combining their own experimental conditions.
  4. Accuracy and uniformity of Tables and Legends.Please carefully check the uniformity and accuracy of the statistical notations (superscript letters) in all tables and figures. 
  5. Please arrange for a thorough proofreading by a native English speaker or a professional academic editing service before the final submission. The format of the [6]reference is incorrect.

Reviewer 3 Report

Comments and Suggestions for Authors

Dear Authors,

I attach a review of the article „Characterization of Coffee Silverskin from Different Origins to Evaluate Its Potential as Ingredient in Novel Food Products”.

The main objective of the present study was to assess the chemical composition, physicochemical, techno-functional, and antioxidant properties of coffee silverskin obtained from Coffea arabica or Coffea canephora of six different origin to determine its potential to be use as an ingredient in the development of novel functional foods. The studies are very interesting especially from the point of view of application perspective, coffee consumption,

properly manage of coffee silverskin as co-product to reduce the environmental impact

and functional food development within circular economy strategies.

However, before publication, the manuscript requires corrections.

Abstract

Line 17-21: … The results revealed an average total dietary fiber content of 75%, with a higher proportion of insoluble fiber compared to soluble fiber. In addition, high levels of phenolic compounds and methylxanthines were identified, including chlorogenic acid, caffeine, and caffeic acid, which showed a significant correlation with the measured antioxidant capacity…

REV: This is already known from many publications. The authors should refer to the parameters for different species of different origins – in fact, they should briefly present the results of their own research. The abstract needs to be revised.

Introduction

Line 30-37: …The Coffee plants are shrubs that belong to Rubiacecae family. There are over 120 30 wild species of coffee plants, but only two of them are cultivated for trade and consump- tion: Coffea arabica (Arabica) and Coffea canephora (Robusta) being C. arabica is the most widely cultivated representing around of 55-60% of the world's coffee supply [1]. The coffee fruits, named cherry, which is approximately 10 mm in size and oval-shaped, contains green coffee beans. These beans are enveloped by several layers: a thin skin referred to as the coffee silverskin, an endocarp known as the parchment, a peptic adhesive layer, the pulp, and the outer skin known as the epicarp [2]...

REV: This obvious information should be removed from the introduction. The authors should focus instead on the problem of using food industry waste as a source of dietary components and biologically active substances. In the introduction, the authors should indicate what new knowledge their research will contribute. It's likely they aren't examining these parameters for the first time, given the recent emergence of food products containing peel.

Line 73: …physicochemical…; see Line 96: …Physico-chemical properties

REV: Should be unified.

Line 74: …Coffea arabica or Coffea canephora

REV: “or” or “and”?

Materials and Methods

Line 89: …The chemical composition (moisture, ash, protein, and dietary fiber) was…

REV: moisture, ash… – it's not a chemical composition – must be corrected.

REV: All parameters should be listed (fat, IDF, SDF).

Line 103: … color coordinates, L* (lightness), a* (redness), and b* (yellowness)…

REV: See Table 3: L*, a*, b*. Italics or not italics? Should be unified and corrected.

Line 108: … Water and oil holding capacity (WHC and OHC, respectively) were determined…

REV: When the first time abbreviations appear, the full name must appear. Should be corrected.

See Lines 299-300 …water-holding capacity (WHC), oil holding capacity (OHC), and swelling capacity (SWC)…

REV: again? Should be corrected.

See Line 313: Water Holding Capacity; B: Oil holding Capacity; C: Swelling capacity; Line 330: …oil holding capacity…

REV: The spelling should be standardized. Should be corrected.

Line 122: … at -40 ºC until…

REV: Should be corrected.

Line 137 and 139

REV: ml or mL? Should be unified and corrected.

Line 153: … ferrous ions chelating activity (FIC)…; see Line 463: …iron-chelating activity (FIC)…

REV: ???

REV: Should be corrected – explanation only for the first time.

Line 161: … (p < 0.05)…; see Line 168: …(p < 0.05)…; Line 287: …(p < 0.05); Line 288: (p <0.05)…

REV: Which version is correct? Should be unified and corrected – italics; spaces.

Line 178: …14.21 g/100 g…; see Line 180: … 20.0 g/ 100 g…

REV: Spaces. Applies to the entire manuscript.

Line 191-192: Table 1

REV: Fat 2.10a and 1.70a very low SD, difference between means near 25% and lack statistically significant differences? Please check. Explanation of TDF, IDF, and SDF should be given.

Line 192-193: …SSFI: Coffee silverskin flour from India; SSFG: Coffee silverskin flour from Guatemala; SSFE: Coffee Silverskin flour from Ethiopia…

See …. Line 382-383: …SSFI: Coffee Silver skin flour from India; SSFG: Coffee silver skin flour from Guatemala….

REV: silverskin or Silverskin? or Silver skin? or silver skin?

REV: Should be unified and corrected.

Line 225: Table 2 ; Line 228-229: …For the same flour, values with different capital letters within the same row indicate significant differences (p < 0.05) according to Tukey´s multiple range test…

REV: Such a comparison has no logical basis. It is impossible to compare the calcium or potassium content, which is naturally several dozen times, or even over 100 times, higher than the iron content. The results of this comparison must be deleted, as well as their description.

Line 244: …the specie C. canephora,…

REV: Should be corrected.

Line 264: …Physicochemical Properties…

REV: Should be corrected.

Line 306: …the specie, C. arabica

REV: Should be corrected.

Line 312: …Figure 1.

…A: Water Holding Capacity; B: Oil holding Capacity; C: Swelling capacity…

REV: Should be corrected.

REV: “B” should be corrected – centred.

REV: Different spelling of the acronyms in the figure and in the caption, e.g. SSF I and SSFI.

Line 323: …from Arabica specie…

REV: Should be corrected.

Line 332: …to specie C. canephora

REV: Should be corrected.

Line 337: …from arabica specie…

REV: Arabica or arabica? Should be corrected.

Line 342: …SSFG. SSFE…

REV: Should be corrected.

Line 382: …SSFI: Coffee Silver skin flour from India; SSFG: Coffee silver skin flour from Guatemala…

REV: Silver or silver? Should be unified corrected.

Line 481: …the DPPH- IC…

REV: space

Line 484: …C. Arabica

REV: Should be corrected.

In the presented papers (Results and Discussion) authors presented physico-chemical properties of the flours obtained  from coffee silverskin of different origins. For each feature, the authors cite results from other authors, indicating that this issue has been extensively studied. However, the question of why the authors compared material of different origins remains unanswered. Were these conditions related to climate, soil, or perhaps processing technology contributed to such significant variation in the parameters studied? Differences were shown and what results from it? The discussion must be supplemented with this issue.

Conclucion

The summary is very general. The authors did not present conclusions based on a comparison of six coffee varieties. It is worth proposing one of the tested types as particularly valuable in the perspective of use in the food industry. Which type has the greatest potential to be use as an ingredient in the development of novel functional foods? The cited literature on the researched issues is very rich – what new things have the authors presented for science?

This section must be improved.

References

This section must be prepared according to the publisher's requirements.

REV: Why are some titles in capital letters and some in small letters?

See Lines 534-535: Cookies enriched with coffee silverskin powder and coffee silverskin ultrasound extract to enhance fiber content and antioxidant properties… and Line 555: …Coffee Silver Skin: Chemical Characterization with Special Consideration of Dietary Fiber and Heat-Induced Contaminants…

Should be unified and corrected – applies to the entire section.

Line 547: …Statista. https://www.statista.com/topics/1856/coffee/ …

REV: underlined?

Line 548: …Nzekoue, F. K….

REV: Space yes or no? Should be unified and corrected – applies to the entire section.

Line 566: …Citrus Sinensis L. Cv. Liucheng….

REV: All Latin names of species should be written in italics. Should be unified and corrected – applies to the entire section.

Line 580: …In Vitro…

REV: Should be Italics. Applies to the entire manuscript.

Line 592: …M.D.d….

REV: ?

Line 597: …& Álvarez, C. A…

REV: Different form? Applies to the entire manuscript.

Line 602: …2830–2843…;

Line 608: …11:1510564….;

Line 636: J.Food Sci. 89(10),6098-6112….;

Line 654: …155–165….;

Line 662: …3), 677-685

REV: Typographic characters should be standardized; spaces, dots …. . Applies to the entire manuscript.

Reviewer 4 Report

Comments and Suggestions for Authors

The manuscript presents relevant and timely research on coffee silverskin; however, the novelty and contribution of the study are not yet sufficiently evident in its current form. Strengthening the Introduction and Discussion, particularly by more clearly articulating the research gap and positioning the findings in relation to existing literature, would help enhance the interest and impact of the work. The following comments provide specific suggestions for improvement.

Comment 1:

It would be beneficial for the authors to explicitly state the research gap in the Introduction. While previous studies on coffee silverskin are mentioned, the specific limitations of the existing literature and how the present study addresses these gaps should be more clearly articulated to better highlight the novelty and significance of the work.

Comment 2:

Given that the study analyzes both free and bound polyphenolic compounds, the Introduction would benefit from a clearer explanation of these two fractions and their importance, particularly in relation to bioavailability and functionality.

Comment 3:

It would be valuable to further discuss the potential of coffee silverskin as a functional ingredient in novel food products, particularly within the framework of sustainability and the circular economy in the food industry. In addition, a more comprehensive discussion comparing the overall properties with existing literature, together with a clearer explanation of the potential contribution to the circular economy, would enhance the impact of the manuscript.

Comment 4:

The authors should provide a more detailed explanation of the model or the samples of novel functional foods derived from coffee silverskin.

Comment 5:

While the topic of the study is relevant, the current presentation of the experimental design and results appears to provide a somewhat limited perspective. Strengthening the experimental framework and more clearly positioning the findings in relation to existing studies may help to better highlight the contribution and originality of the work.

Comment 6:

The results section presents useful data; however, the overall dataset appears rather limited. Providing additional data, such as a more comprehensive profile of other phenolic compounds, and comparing these findings with previous study.

Reviewer 5 Report

Comments and Suggestions for Authors

The title addresses a relevant and timely topic, as coffee silverskin is an important agri-food by-product with growing interest for sustainable food applications:

  • The extraction procedure is generally well described, but some details are missing or unclear:

    • Storage at 40 °C appears unusually high for antioxidant extracts and may cause degradation; please justify this condition
    • Please ensure consistent formatting (e.g., “EDTA was used” instead of “EDTAwas

      Interpretation of free vs. bound polyphenols

      • The observation that bound polyphenols predominate over free polyphenols in all samples is important and should be discussed more explicitly in relation to:

        • Cell wall association of phenolics

        • The effect of roasting on phenolic binding and release

        • Implications for bioaccessibility and functionality

        • While the chemical characterization of polyphenolic compounds provides valuable insight into the bioactive potential of coffee silverskin, the manuscript would be significantly strengthened by the inclusion of a sensory perception evaluation. Sensory analysis would help to relate the compositional differences observed among samples from different geographical origins to perceived attributes such as bitterness, astringency, aroma, and overall acceptability.

          In particular, correlating key phenolic compounds (e.g., caffeoylquinic acids and ferulic acid derivatives) with sensory responses could provide a more comprehensive understanding of the functional relevance and practical applicability of coffee silverskin as a food ingredient. If sensory analysis cannot be included experimentally, a more detailed discussion of potential sensory implications based on existing literature is strongly recommended.

Comments on the Quality of English Language

not bad

Round 2

Reviewer 1 Report

Comments and Suggestions for Authors

I thank the authors for their responses and the improvements made. I feel that the manuscript is ready for publiacation now.

Reviewer 3 Report

Comments and Suggestions for Authors

Accept in present form

Reviewer 4 Report

Comments and Suggestions for Authors

The authors have carefully and thoroughly revised the manuscript in response to all of my comments. The implemented changes have notably enhanced the quality and readability of the work, making the manuscript more engaging and impactful.
I have no further comments and recommend the manuscript for publication.

Reviewer 5 Report

Comments and Suggestions for Authors

accept

Comments on the Quality of English Language

accept